# Inhibition of MACC1-Induced Metastasis in Esophageal and Gastric Adenocarcinomas

**DOI:** 10.3390/cancers14071773

**Published:** 2022-03-31

**Authors:** Christoph Treese, Jessica Werchan, Moritz von Winterfeld, Erika Berg, Michael Hummel, Lena Timm, Beate Rau, Ole Daberkow, Wolfgang Walther, Severin Daum, Dennis Kobelt, Ulrike Stein

**Affiliations:** 1Experimental and Clinical Research Center, Charité–Universitätsmedizin and Max-Delbrück-Center for Molecular Medicine in the Helmholtz Association, 13125 Berlin, Germany; christoph.treese@charite.de (C.T.); jessica.werchan@helios-health.com (J.W.); lena.timm@mdc-berlin.de (L.T.); wowalt@mdc-berlin.de (W.W.); dennis.kobelt@mdc-berlin.de (D.K.); 2Department of Gastroenterology, Infectious Diseases and Rheumatology, Campus Benjamin Franklin, Charité–Universitätsmedizin Berlin, Corporate Member Freie Universität Berlin and Humboldt-Universität zu Berlin, 12203 Berlin, Germany; severin.daum@charite.de; 3Berlin Institute of Health (BIH), 10178 Berlin, Germany; 4Institute of Pathology, Charité–Universitätsmedizin Berlin, Corporate Member of Freie Universität Berlin and Humboldt-Universität zu Berlin, 10117 Berlin, Germany; m.vonwinterfeld@pathologie-rosenheim.de (M.v.W.); erika.berg@charite.de (E.B.); michael.hummel@charite.de (M.H.); 5Department of Surgery, Campus Virchow-Klinikum and Campus Mitte, Charité–Universitätsmedizin Berlin, Corporate Member of Freie Universität Berlin and Humboldt-Universität zu Berlin, 10117 Berlin, Germany; beate.rau@charite.de; 6EPO Experimentelle Pharmakologie & Onkologie, 13125 Berlin, Germany; ole.daberkow@epo-berlin.com; 7German Cancer Consortium (DKTK), 69120 Heidelberg, Germany

**Keywords:** gastric cancer, esophageal cancer, metastasis, MACC1, MEK1 inhibition, selumetinib

## Abstract

**Simple Summary:**

Esophageal and Gastric Adenocarcinomas (AGE/S) are characterized by early metastasis and poor survival. MACC1 (Metastasis Associated in Colon Cancer 1) acts in colon cancer as a metastasis inducer and is linked to reduced survival. In this study, we analyzed the prognostic role of MACC1 in a large AGE/S cohort and the potential of MACC1 inhibition in vitro and in vivo. MACC1 is an independent negative prognostic marker in our cohort. In vitro, migration was enhanced by MACC1 in overexpressing cells. This MACC1-related effect could be inhibited by using selumetinib in vitro. In vivo, MACC1 induced faster and larger metastasis development, which could be inhibited by selumetinib. In conclusion, MACC1 is a strong negative prognostic factor in AGE/S and is a potential target for therapy with selumetinib.

**Abstract:**

Esophageal and Gastric Adenocarcinomas (AGE/S) are characterized by early metastasis and poor survival. MACC1 (Metastasis Associated in Colon Cancer 1) acts in colon cancer as a metastasis inducer and is linked to reduced survival. This project illuminates the role and potential for the inhibition of MACC1 in AGE/S. Using 266 of 360 TMAs and survival data of AGE/S patients, we confirm the value of MACC1 as an independent negative prognostic marker in AGE/S patients. MACC1 gene expression is correlated with survival and morphological characteristics. In vitro analysis of lentivirally MACC1-manipulated subclones of FLO-1 and OE33 showed enhanced migration induced by MACC1 in both cell line models, which could be inhibited by the MEK1 inhibitor selumetinib. In vivo, the efficacy of selumetinib on tumor growths and metastases of MACC1-overexpressing FLO-1 cells xenografted intrasplenically in NOG mice was tested. Mice with high-MACC1-expressing cells developed faster and larger distant metastases. Treatment with selumetinib led to a significant reduction in metastasis exclusively in the MACC1-positive xenografts. MACC1 is an enhancer of tumor aggressiveness and a predictor of poor survival in AGE/S. This effect can be inhibited by selumetinib.

## 1. Introduction

Esophageal and Gastric Adenocarcinomas (AGE/S) were responsible for over one million new cancer cases and an estimated 769,000 deaths in 2020 worldwide [1]. The asymptomatic development in early tumor stages and the rapid transformation in advanced tumor stages with reduced possibilities of being cured pose a concealed risk for the patient and a diagnostic challenge to clinicians. In cases of symptomatic disease, lymph nodes and metastases are already present in about 50% and 36% of cases, respectively [1,2,3]. This early spread is mainly responsible for the high disease burden and leads to death within a short time [4]. 

From our point of view, the prognosis of AGE/S patients can be improved by two factors: First, the development of prognostic biomarkers that are able to identify and discriminate among patients with a high and low metastasis potential, to stratify the aggressivity of therapy and reduce undue therapeutic side effects. Second, but even more important, the development of new targeted therapeutic strategies.

In 2009, we identified the gene MACC1 (Metastasis Associated in Colon Cancer 1) in human colorectal cancer [5]. MACC1 induces cell proliferation, dissemination, migration, and invasion in vitro, as well as tumor progression and metastasis formation in vivo [5,6,7,8]. Detection of high levels of MACC1 transcripts in colorectal cancer tumor tissue or blood samples is predictive for metastasis formation and shorter overall survival [5,8,9]. Beside colorectal cancer, MACC1 has been confirmed as a prognostic biomarker in a variety of other solid cancers such as pancreatic, hepatocellular/biliary, lung, ovarian, breast, renal, bladder, nasopharyngeal cancer, glioblastoma, and osteosarcoma [10]. The role of MACC1 in gastric cancer was previously analyzed exclusively in studies with Asian patients, where MACC1 was shown to be a negative prognostic marker and associated with the presence of distant metastasis [11,12,13,14,15,16,17,18]. However, these studies had short follow up periods, and none analyzed the effect of MACC1 in a multivariate analysis. Beside the prognostic role of MACC1 in several tumor entities, it harbors the potential as a druggable target in cancer therapy. We could show by in silico and functional analyses that MACC1 is post-translationally activated by phosphorylation attained by mitogen-activated protein kinase 1 (MAP2K1, MEK1). The MACC1-induced biological and phenotypical effects were reduced in vitro and in vivo by using the MEK1-inhibitor selumetinib (AZD6244) in colorectal cancer [19].

In this study, we first clarify the important prognostic role of MACC1 in a large Caucasian AGE/S cohort with a long follow-up period. More importantly, however, we show the strong potential of MACC1 as a therapeutical target of selumetinib (AZD6244) in AGE/S by using MACC1 knockdown and MACC1 overexpressing cell line models in vitro and a MACC1 overexpressing cell line xenograft model in vivo.

## 2. Materials and Methods

### 2.1. Patients

Clinical data from patients with AGE/S of all tumor stages, primarily treated by surgery between 1992 and 2004 at the Charité–Universitätsmedizin Berlin, were collected retrospectively. The mean follow-up was 121.7 months (95% CI: 113.9–129.5). Overall survival, used as a measure of prognosis, was defined as the time from diagnosis to death or the last follow-up. Disease-specific survival was defined as the time from diagnosis to tumor-related death or the last follow-up. The data, including patient characteristics and follow-up information, were retrieved from the patient management software (SAP^®^) and the regional population-based cancer registry (“Gemeinsames Krebsregister”) and are summarized in Table 1. This study was approved by the Institutional Review Board of the Charité (EA4/115/10).

### 2.2. Tissue Samples

Tissue samples were collected from the archive of the Department of Pathology, Charité–Universitätsmedizin Medicine Berlin. Paraffin-embedded tumor samples (*n* = 360) were available from surgically treated chemotherapy-naive patients. All samples were reevaluated according to histological diagnosis, tumor stage, and grade, and classified by the histological architecture of AGE/S carcinoma using Lauren and Ming classification by a specialist in gastrointestinal pathology. Data concerning tumor size, depth of invasion, and tumor invasion of veins or lymphatic vessels were retrieved from the Charité–Universitätsmedizin Berlin patient management software. Tissue samples were screened in a hematoxylin-eosin (HE)-stained section for representative areas of the center of the solid tumors. Two 1 mm-diameter tissue cores were punched out from each of the 360 available cases and were transferred to a recipient paraffin block. After re-melting, sections (4 µm thick) were consecutively cut from each tissue microarray block. HE-staining was performed on tissue micro array (TMA) sections for reconfirmation of the content of the tumor and non-tumor tissue in each core. Immunohistochemical analysis was performed on TMA sections using a MACC1-specific monoclonal antibody (Sigma^®^, Burlington, MA, USA; Clone HPA020103). After pretreatment with 0.001 N citrate buffer (pH = 6.0), the protein kinase K, and hydrogen peroxide, the primary antibody was used in a 1:100 dilution. As the secondary antibody, an anti-rabbit horse radish peroxidase (HRP) bound antibody (Promega^®^, Fitchburg, MA, US was used in a 1:200 dilution. The staining was finalized using a 3,3-diaminobenzidine (DAB) peroxidase (HRP) substrate kit (Vector Laboratories^®^, Burlingame, CA. USA.

MACC1 expression was evaluated by an immunoreactivity score (IRS) percentage of the stained tumor cells (0 = 0%, 1 = 1–25%, 2 = 26–50%, 3 = 51–75%, 4 = 76–100%), multiplied with the staining intensity (score 0–3 = no staining to strong staining) to give the IRS score of each sample (score 0–12). To define the cut-off value, an ROC curve analysis was performed. The IRS cut-off value 5 reaches a sensitivity of 76.6% and a specificity of 64.2% (Youden Index. 0.408) and was chosen as the cut-off value (see Appendix A). Samples with IRS > 5 were assessed as MACC1-“high” tumors, and samples with <5 as MACC1-“low” tumors (Figure 1).

For the mismatch-repair (MMR) status, the immunohistochemistry was performed on an automated staining system (BenchMark Ultra, Roche Ventana, Mannheim, Germany) using the prediluted antibodies (all Roche Ventana) MLH1 (clone M1), PMS2 (clone EPR3947), MSH2 (clone G219-1129), and MSH6 (clone 44), following the manufacturer’s instructions.

### 2.3. Cell Lines and Medium

The human AGE/S cell lines OE33 (Caucasian), MKN45 (Asian), NCI-N87 (Caucasian), OAC-P4C (Caucasian), and FLO-1 (Caucasian) were maintained in RPMI 1640 (PAA Laboratories GmbH, Coelbe, Germany) supplemented with 10% fetal bovine serum (FBS) (PAA Laboratories GmbH, Coelbe, Germany) in a humidified incubator at 37 °C and 5% CO_2_. Cells were analyzed for potential contamination and were found to be free of mycoplasma contamination. Authentication was performed by short tandem repeat genotyping at the DSMZ (German Collection of Microorganisms and Cell Cultures; Braunschweig, Germany) and was consistent with the published genotypes for these cell lines.

### 2.4. Cloning and Transduction

The lentivirus vector coding for MACC1RFP was generated by replacing GFP in pLenti-MACC1GFP (Origene) by tagRFP (GenBank ABR08320). The template DNA for cloning was synthesized by Integrated DNA Technologies (IDT, Coralville, IA, USA) as a gene block fragment. Only the control plasmid encoding for tagRFP was generated by removing the MACC1 cDNA. The vector pLenti-eGFPLuc was generated by replacing MACC1GFP in pLenti-MACC1GFP by eGFPLuc taken from pEGFPLuc (Clontech/Takara, Mountain View, CA, USA). The control vector was generated by removing the luciferase cDNA. Lentiviruses were generated by a three-plasmid transfection of HEK293T cells using psPAX2 (Addgene Plasmid #12260) and pMD2.G (Addgene Plasmid #12259) for packaging and VSV-G pseudotyping. Target cells were transduced with a MOI of 10 without additives. Positive cells were selected via FACS for GFP (eGFP, eGFPLuc) and RFP (RFP, MACC1RFP).

### 2.5. MACC1 Expression Analysis

For RNA expression analysis, 3 × 10^5^ cells were seeded in a 6-well plate and total RNA was isolated using the Universal RNA Purification Kit (Roboklon, Berlin, Germany) according to the manufacturer’s instructions. RNA was quantified (Nanodrop, Peqlab, Erlangen, Germany), and 50 ng of RNA was reverse transcribed with random hexamers in a reaction mix (10 mM MgCl_2_, 1× RT-buffer, 250 μM pooled dNTPs, 1 U/μL RNAse inhibitor, and 2.5 U/μL Moloney Murine Leukemia Virus reverse transcriptase; all from Thermo Fisher Scientific, Waltham, MA, USA) at 42 °C for 15 min and 99 °C for 5 min, with subsequent cooling to 5 °C for 5 min. The cDNA was amplified by quantitative polymerase chain reaction (qPCR) using SYBR Green dye chemistry and the LightCycler 480 (Roche Diagnostics, Mannheim, Germany) under the following PCR conditions: 95 °C for 2 min followed by 45 cycles of 95 °C for 7 s, 60 °C for 10 s, and 72 °C for 20 s using primers for MACC1 and G6PD as described previously [5]. The same protocol for qPCR had been employed for RNA from shock-frozen tumor and liver tissue samples from animals. Human satellite DNA in the liver sections of the control and treated mice was determined as previously described [8]. Data analysis was performed with the LightCycler 480 Software release 1.5.0 SP3 (Roche Diagnostics). Mean values were calculated from duplicate qRT-PCR reactions. Each mean value of the expressed gene was normalized to the respective mean of the G6PD cDNA amount. 

For total protein extraction, 3 × 10^5^ cells were placed in 6-well plates. After 24 h, the cells were lysed with RIPA buffer (50 mM Tris–HCl; pH 7.5, 150 mM NaCl, and 1% Nonidet P-40, supplemented with complete protease inhibitor tablets; Roche Diagnostics) for 30 min on ice. Protein concentration was quantified with Bicinchoninic Acid Protein Assay Reagent (Thermo Fisher Scientific) according to the manufacturer’s instructions. Lysates of equal protein concentration were separated with sodium dodecyl sulfate-polyacrylamide gel electrophoresis (SDS-PAGE) and transferred to Hybond-C Extra nitrocellulose membranes (GE Healthcare, Munich, Germany). Membranes were blocked for 1 h at room temperature with 5% nonfat dry milk in TBST buffer (10 mM Tris-HCl; pH 8, 0.1% Tween 20, and 150 mM NaCl). Membranes were then incubated overnight at 4 °C with MACC1 antibody (Sigma-Aldrich, St. Louis, MI, USA, dilution 1:1000) or β-actin antibody (Sigma-Aldrich, dilution 1:10,000), followed by incubation for 1 h at room temperature with HRP-conjugated anti-rabbit IgG (Promega, dilution 1:10,000) or anti-mouse IgG (Thermo Fisher Scientific, dilution 1:10,000). Antibody-protein complexes were visualized with WesternBright ECL HRP substrate (Advansta, Menlo Park, CA, USA) and subsequent exposure to CL-Xposure Films (Thermo Fisher Scientific). Immunoblotting for β-actin served as the protein-loading control. 

### 2.6. Proliferation

Proliferation was analyzed in real-time using a xCELLigence system E-Plate (Roche^®^). 3000 FLO-1 or OE33 cells were seeded with fresh RPMI medium in each well. The impedance value of each well was automatically monitored every 2 h by the xCELLigence system for a duration of 24 h and expressed as a CI (cell index) value. After normalization, the area under the curve over 24 h (AUC-24 h) was calculated to compare proliferation between the different cell lines. 

Additionally, proliferation for FLO-1/EV and FLO-1/MACC1 was analyzed over 72 h by MTT. 4 × 10^3^ cells were plated into 96-well plates and were allowed to accommodate for 24 h. After 48 and 72 h, formazan crystals were dissolved in 150 μL of DMSO and the absorption was measured at 560 nm in the absorbance reader (Tecan infinite 200 PRO, Tecan, Männedorf, Switzerland). Each cell proliferation experiment was performed in triplicates. 

### 2.7. Migration

For migration analysis, the chemotaxis module of the live-cell analysis system IncuCyte ZOOM^®^ was used. 1000 cells were seeded with 60 µL RPMI + 0.1% FBS in the insert of an IncuCyte ClearView 96-well cell migration plate. The lower compartments of the chamber were filled with 160 µL RMPI + 10% FCS. Images of each insert were taken every 2 h for a period of 72 h. The migration from the top to the bottom side of the membrane was quantified as the migrated cells on the bottom of the membrane in relation to the total cell number added.

### 2.8. In Vitro Drug Treatment

Selumetinib (AZD6244 GSK1120212: Selleck Chemicals, Munich, Germany) was solubilized in DMSO and was added in a concentration of 10 µM to the cell medium. As described before, 1000 cells were seeded in the migration plate with cell medium containing selumetinib or DMSO as the negative control. Afterwards migration was analyzed every 2 h for a period of 72 h. 

### 2.9. Cell-Line-Derived Xenograft Model

Experiments were performed in accordance with the United Kingdom Coordinated Committee on Cancer Research guidelines and approved by the responsible local authorities (Reg 0393/17, State Office of Health and Social Affairs, Berlin, Germany). For analysis of MACC1-induced metastasis and the potential for its inhibition with selumetinib in vivo, the MACC1 overexpression cell line FLO-1/MACC1 (MACC1RFP) was used, and the cell line FLO-1/EV (RFP) was used as the control. To monitor metastasis development in vivo, cell lines were transduced with lentiviruses coding for firefly luciferase reporter (pLenti-eGFPLuc, pLenti from Origene) generating FLO 1/MACC1/Luc and FLO-1/EV/Luc. NOG mice were intrasplenically injected with 1 × 10^6^ FLO 1/EV/Luc or 1 × 10^6^ FLO-1/MACC1/Luc cells. Bioluminescence imaging (NightOwl LB 981 systems, Berthold Technologies, Bad Wildbad, Germany) was performed after anesthesia with isofluran (Abbott GmbH, Wiesbaden, Germany) using 150 mg/kg D-luciferin (Biosynth, Staad, Switzerland) in PBS. Tumor growth and metastasis were quantified with ImageJ 1.48 k (NIH, Bethesda, MD, USA). For in vivo inhibition of MEK1, animals were treated twice a day with 2 mL/kg solvent (10% Kolliphor EL, 0.9% NaCl) or 50 mg/kg selumetinib orally twice a day. Treatments started at the day of transplantation and were continued until the animals were euthanized. Due to ethical reasons and in accordance with the local authorities, the animal experiments were terminated by cervical dislocation. Livers were collected and snap frozen for molecular analysis. For detection of metastasized human cells in mouse livers, genomic DNA was isolated from mouse livers (DNA-RNA-Protein Extraction Kit, Roboklon, Berlin, Germany). Quantitative PCR was performed using 50 ng genomic DNA. Titration was performed with genomic DNA from spiked human/mouse cell populations. Primer sequences for human satellite DNA (BioTeZ and TIB MolBiol, Berlin, Germany) were used as described previously [20]. In parallel, human CK19 was detected by immunohistochemical staining using an anti-CK19 antibody (TA336845, OriGene, Rockville, MD, USA, 1:200). CK19 protein was visualized using Dako DAB liquid (Agilent).

### 2.10. Statistics

Statistical analysis was performed using IBM SPSS Version 24 and GraphPad prism version 8 (GraphPad Software, San Diego, CA, USA). Overall survival was evaluated in months from the time of diagnosis until death or until the most recent follow-up using Kaplan–Meier plots. Associations of MACC1 expression with tumor size, distant and lymph node metastasis, venous and lymphatic infiltration, Lauren and Ming classification, and UICC grading and classification were tested by the X^2^ test. Univariate survival analyses were performed according to the Kaplan–Meier method, using the log-rank test for assessments of statistical significance. Cox’s regression was used for multivariate analysis in a stepwise backwards procedure with the level of significance set to *p* < 0.05. A *p*-value of < 0.05 was considered statistically significant. Experimental in vitro and in vivo data were analyzed by two-sided t-tests (two groups) or ANOVA (more than two groups).

## 3. Results

### 3.1. Patients

Data of 360 patients (detailed clinicopathological characteristics are summarized in Table 1) were analyzed in this study (female = 131, males = 229, median age = 62.06 years). In 300 cases (83.3%), the tumor was localized in the stomach, and in 60 cases (16.7%), in the esophagus or gastroesophageal junction. Patients with all tumor stages (T1 = 47, T2 = 140, T3 = 122, T4 = 32, Tx = 19) and all nodal (N0 = 105, N+ = 254, Nx = 1) and metastasis statuses (M0 = 245, M1 = 115) were included. Data on lymphatic infiltration were available in 321 cases, and data on venous infiltration in 315 cases. Lymphatic infiltration was observed in 206 patients (57.2%), and venous infiltration in 114 patients (31.7%). The mean follow-up was 121.7 months (95% CI: 113.9–129.5). The 5-year overall survival was 38.1%, and the 5-year disease-specific survival was 45.4%.

### 3.2. MACC1 Expression

Due to the procedure of cutting and staining, some samples could not be used for the evaluation. Samples with insufficiently representative tumor tissue were also excluded from the evaluation. Cores with representative tumor material and evaluable staining were available in 266 of 360 samples (73.9%), with 211 samples (79.3%) being MACC1-positive (samples with IRS > 5) and 55 samples (20.7%) being MACC1-negative (samples with IRS < 5) (Figure 1).

The correlation of MACC1 expression status and patient characteristics showed significantly more MACC1-high patients with higher T-stages (*p* = 0.001), in M1- (*p* = 0.012) and G3-staged (*p* = 0.001) diseases, and with MMR-deficient tumors (*p* < 0.0001) (see Table 1). Patients with MACC1-high tumors showed a median survival of 54.2 months (95% CI 45.0–63.4), which is significantly shorter compared to the 87.1 months (95% CI: 66.5–107.6, *p* = 0.002) for MACC1-low patients (see Table 2 and Figure 2A). The disease-specific survival was 98.8 months (95% CI: 77.1–120.5) for MACC1-low and 68.8 months (95% CI: 575–80.2; *p* = 0.011) for MACC1-high patients (see Figure 2B). The impact of MACC1 as a negative prognostic factor was significant (OR 1.51 [95% CI: 1.013–2.258]; *p* = 0.043) in a multivariate Cox’s regression model (Table 2). In subgroup analysis (vein invasion and lymphatic vessel invasion), MACC1 was a strong predictor of reduced survival in tumors with a low risk morphology: V0 tumors: MACC1-low, 115.7 months (95% CI: 92.0–139.3) vs. MACC1-high, 65.4 months (95% CI: 52.5–78.3; *p* < 0.001) (Figure 2C); L0 tumors: MACC1-low, 134.9 months (95% CI: 107.0–162.7) vs. MACC1-high, 80.9 months (95% CI: 63.0–98.8; *p* = 0.003) (Figure 2D).

### 3.3. In Vitro Analysis

The AGE/S cell lines NCI-N87, FLO-1, OE33, and OACP4C were screened for MACC1 expression by RT-PCR and Western blot analysis. Out of these, FLO-1 was identified as a cell line with virtually no MACC1 expression (ratio 0.01 RNA MACC1/G6PDH) and OE33 as a cell line with a high endogenous MACC1 expression (ratio 0.75 RNA MACC1/G6PDH) (Figure 3A). From FLO-1 cells, lentiviral MACC1 overexpressing cells (FLO-1/MACC1) were generated with a high MACC1 expression (ratio 6.29 RNA MACC1/G6PDH) compared to the wild type and lentiviral control clone (FLO-1/EV) (Ratio 0.01 RNA MACC1/G6PDH) (Figure 3B).

For OE33 cells, lentiviral MACC1 knockdown clones and control clones with scrambled SH vectors were generated. From the MACC1 knockdown clones, the clone with the lowest MACC1 expression (ratio 0.34 RNA MACC1/G6PDH) was chosen for further analysis (OE33/shMACC1-). The control clone (OE33/shCTRL) showed higher MACC1 expression (ratio 1.76 RNA MACC1/G6PDH) compared to the wildtype (ratio 0.75 RNA MACC1/G6PDH) (Figure 3C). There was a limited degree of proliferation change after stable knockdown (−12.9%) or overexpression (+18.1%) of MACC1 in OE33 or FLO-1, respectively (Figure 3D). The additional analysis of the proliferation over 72 h showed no significant differences between FLO-1/EV (100% ± 4.886) and FLO-1/MACC1 (112.3% ± 12.24; *p* = 0.113) (see Appendix A).

In contrast to cell proliferation, migration analysis showed a significant two-fold increase (208.60%) in FLO-1/MACC1 cells compared to FLO-1/EV cells (*p* = 0.004) and a significant, nearly two-fold decrease in MACC1 knockdown clones OE33/shMACC1 (58.81%; *p* = 0.013) compared to OE33/shCTRL (Figure 3E,F).

After the establishment of the MACC1 overexpression cell line FLO-1/MACC1 and the knockdown cell line OE33/shMACC1, and significant detection of MACC1-modulated migration, the cell lines were treated with the MEK1 inhibitor selumetinib. The treatment with selumetinib led to a significant reduction in the migratory ability of FLO-1/MACC1 cells down to 32.23% (*p* = 0.0001) compared to the untreated FLO-1/MACC1 (Figure 3G). In the OE33/shMACC1 cells, migration was marginally reduced (84.94%; *p* = 0.253) compared to the untreated cells. In endogenously MACC1-expressing OE33/shCTRL cells, however, selumetinib had a significant decreasing effect on migration (18.68% *p* = 0.001) (Figure 3H,I).

### 3.4. In Vivo Analysis

The in vitro analysis showed a strong effect of MACC1 on the acceleration of migration. Based on our previous experience, we chose for our in vivo analysis a xenograft model in which the cells are transplanted intrasplenically [8]. This model makes it possible to observe not only the local proliferation but also the metastasis formation of human cells into the mouse liver. To evaluate the impact of MACC1-induced metastasis, the AGE/S cell line FLO-1 derived model was used.

Six mice were xenotransplanted by intrasplenic injection of FLO-1 cells stably transduced with EV or MACC1. At the experimental endpoint (tumor burden), the signals in the livers of FLO-1/MACC1/Luc-cell-transplanted animals (6.48 × 10^8^ RLU) exceeded the signals of the control animals (3.29 × 10^8^ RLU, *p* = 0.0823) by nearly two-fold (Figure 4A). A mixed model two-way ANOVA revealed a strongly significant influence of time (*p* < 0.0001) and a statistically significant influence of MACC1 (*p* = 0.0305) on metastasis formation in this model over time. This also confirms the metastasis-promoting abilities of MACC1 for gastric cancer.

Next, we used this model to proof for metastasis inhibition of FLO-1/MACC1/Luc-cell-transplanted tumors by selumetinib. Again, NOG animals were xenotransplanted with FLO 1/EV/Luc or FLO-1/MACC1/Luc cells. Starting on the day of transplantation, the animals were treated orally twice daily with 50 mg/kg of selumetinib or solvent for three weeks. At the experimental endpoint (tumor burden), the luminescence as a measure of metastasis load was analyzed (Figure 4B). Again, we observed an enhanced metastasis formation by MACC1 in animals treated only with the solvent (EV: 5.91 × 10^8^ RLU vs. MACC1: 9.62 × 10^8^ RLU). The treatment with selumetinib, however, led to a moderate metastasis reduction in the FLO 1/EV/Luc control cells by 30% (EV vehicle treated 5.91 × 10^8^ RLU vs. EV selumetinib treated 4.15 × 10^8^ RLU). In contrast, selumetinib reduced the metastasis burden of FLO 1/MACC1/Luc-xenotransplanted animals by 3-fold as measured by in vivo bioluminescence imaging (MACC1 vehicle treated 9.62 × 10^8^ RLU vs. MACC1 selumetinib treated 3.00 × 10^8^ RLU). To additionally quantify the metastasis burden in these animals, the liver tissue of euthanized animals was collected and snap frozen. Using a quantitative PCR test for human satellite DNA, we showed a significant increase in human cells in the murine liver of FLO-1/MACC1/Luc-cell-transplanted mice compared to control mice by 42% (EV vehicle vs. MACC1 vehicle *p* = 0.023). Treatment with selumetinib had no effect on FLO1/EV cells compared to vehicle-treated cells. Using FLO1/MACC1/Luc cells, selumetinib treatment significantly decreased metastasis formation by two-fold compared to FLO1/EV/Luc (*p* = 0.04), and three-fold compared to vehicle treated animals xenografted with FLO1/MACC1/Luc cells (*p* < 0.0001, Figure 4C). This was further supported by staining of human CK19 as a marker for metastasized cells in the murine liver (Figure 4D).

In summary, the results showed that selumetinib treatment inhibited MACC1-induced metastasis formation in the AGE/S FLO-1 tumor model in vivo.

## 4. Discussion

This study shows a strong correlation of MACC1 expression with poor overall survival in a large Caucasian gastric cancer and esophageal adenocarcinoma cohort. As the main finding, we demonstrated that the MEK1 inhibitor selumetinib is an effective inhibitor for MACC1-induced migration in vitro and MACC1-induced metastasis in vivo in AGE/S. These results offer a new treatment possibility for a large group of high-risk AGE/S patients.

Before this study, the prognostic role of MACC1 in gastric cancer had been analyzed in other studies with a sample size between 98 and 436 patients, confirming the negative prognostic role of MACC1 [13,14,16,17,18]. [A meta-analysis from 2019 showed, in a total number of 2103 gastric cancer patients, a significant correlation of MACC1 expression levels in tumor tissue with distant metastasis and vascular invasion in gastric cancer patients [21]. Beside these studies on tumor tissue, in 2015 our research group published an analysis of the levels of circulating MACC1 transcripts in the plasma of a small Caucasian gastric cancer patients cohort. We were able to show that detection of MACC1 transcripts were of diagnostic value and were prognostic for patient survival [22].

It is remarkable that all studies on tumor tissue were conducted in Asian patient populations and that there are no data from Caucasian populations. Several studies have found substantial differences between Asian and Caucasian gastric cancer patients. These affect the incidence and prognosis of gastric cancer, as well as its genetic and biological properties [1,23,24,25,26]. The most prominent example of biology operating differently across ethnic groups was the AVAGAST trial, analyzing the effect of bevacizumab in gastric cancer: only the non-Asian subgroup showed a benefit from bevacizumab in combination with chemotherapy [27].

This present study demonstrates the negative prognostic role of MACC1 for the first time in a large Caucasian AGE/S population. With 79.3% MACC1-positive tumor samples, our cohort has a higher rate of MACC1-expressing samples compared to the studies of Asian populations (mean 57.8% (34.9–80.6)). Regarding the histomorphological characteristics, our data are comparable with those of the studies of Asian populations. There is also a strong correlation with advanced tumor diseases (T, N, M stage) and the invasiveness of the disease (V and L status) [11,12,13,14,15,16,17,18,21,28].

However, our subgroup analysis shows that MACC1 expression is of particular interest in situations in which other established markers (V0 and L0) predict a good prognosis: patients with no detectable vascular or lymphatic invasion (V0 or L0 situation) showed a significant reduced overall survival when MACC1 was highly expressed (V0: MACC1-low, 115.7 months vs. MACC1-high, 65.4 months; L0: MACC1-low, 134.9 months vs. MACC1-high, 80.9 months). This aspect highlights the potential of MACC1 as a biomarker in clinical practice. Patients with hitherto unknown risk constellations could be identified and treated according to their higher risk of metastasis development, for example by adding a multimodal treatment before surgery.

In the in vitro part of our study, we present two AGE/S cell line models with an effectively forced MACC1 overexpression in the originally MACC1 negative cell line FLO-1 (FLO-1/MACC1) and an effective silencing of MACC1 expression in the endogenously MACC1-expressing cell line OE33 (OE33/shMACC1). In both models, MACC1 levels had just a small effect on cell proliferation but a strong effect on cell migration. The effect on migration is in accordance with other studies using MACC1 overexpressing and silenced cell lines of Asian origin, but in contrast to our results, they could also detect a significantly increased cell proliferation effected by MACC1 [13,29].

In vivo, the intrasplenic injection of the MACC1-overexpressing cell line FLO-1/MACC1 and the control cell line FLO-1/EV was carried out in NOG mice. At the experimental endpoint, the FLO-1/MACC1 group showed a significantly increased burden of metastases in the liver, verified by in vivo luminescence intensity and by the post-mortem detection of human satellite DNA. Two other authors analyzed the role of MACC1 in a subcutaneous xenograft model, with a measurable large tumor size at the experimental end in the MACC1 overexpressing group [13,29]. Wang et al. additionally detected a significantly higher tumor dissemination after intravenous injection of MACC1-overexpressing BGC823 cells [13]. To our knowledge, this is the first study using a metastasis-inducing AGE/S xenograft model by intrasplenic cell injection, where the increased metastatic potential of the MACC1-overexpressing FLO-1 cells was clearly demonstrated compared to MACC1 negative FLO-1 cells.

New therapeutic targets are strongly needed in AGE/S treatment. The approval of trastuzumab for Her2Neu-expressing tumors and nivolumab as a checkpoint inhibitor for PD-L1-expressing tumors were big advantages in the treatment of AGE/S in recent years [30,31]. Unfortunately, only a few patients are eligible for these new therapy options: the predictive biomarker Her2Neu is expressed in approximately 18% of AGE/S tumors [32], and about 28% of gastric cancer are eligible for checkpoint inhibition by having a sufficient PD-L1 expression with a combined positive score >5 [33]. In all published patient cohorts, about 60% of the population were MACC1-positive, and in our study, 79% were positive. This profile makes MACC1 interesting as a target with the possibility of therapeutic use in many patients.

Recently, we showed that MEK1 directly phosphorylates MACC1, leading to activation of MACC1-induced migration and metastasis in colorectal cancer [19]. In a next step, we were able to show that the inhibition of MEK1, either by RNAi or by the approved MEK1 inhibitor selumetinib, leads to a reduction in MACC1-induced effects in vitro and in vivo [19].

This is the first study which analyses the translation of pharmacological MACC1 inhibition for another tumor entity, extending our knowledge focused on colorectal cancer. We were able to show the high effectivity of selumetinib in MACC1-expressing tumor cells and gastric cancer cells in vitro and in vivo. So far, there are only a few studies on selumetinib in AGE/S. One in vitro study of gastric cancer cell lines showed that proliferation, VEGF expression, and ERK phosphorylation were suppressed by selumetinib treatment [34]. The only existing clinical data on selumetinib in gastric cancer are from one arm of an umbrella trial (the VIKTORY umbrella trial). In this arm, 25 KRAS mutant, KRAS-amplified, or MEK-signatured gastric cancer patients received second-line chemotherapy with docetaxel and selumetinib. A partial response was seen in seven (28%) [35]. This result is comparable to the other second-line therapies such as paclitaxel and ramucirumab, which had a partial response of 28% [20]. These results show the potential of selumetinib for this tumor entity. A selection for MACC1 positivity was not performed in this trial. Regarding our results, MACC1 might be a good predictor for this therapy.

## 5. Conclusions

In conclusion, we identified the role of MACC1 as a negative prognostic marker in a large Caucasian AGE/S cohort. High expression levels of MACC1 identify patients with a high risk of metastasis, even if they miss well-known morphological risk factors such as venous or lymphatic invasion. Moreover, this study demonstrates for the first time the successful pharmacological inhibition of MACC1 expression and the in vitro and in vivo MACC1-induced effects caused by the FDA-approved drug selumetinib. The potential of MACC1 as therapeutic target for selumetinib treatment requires confirmation in clinical trials.

## Figures and Tables

**Figure 1 cancers-14-01773-f001:**
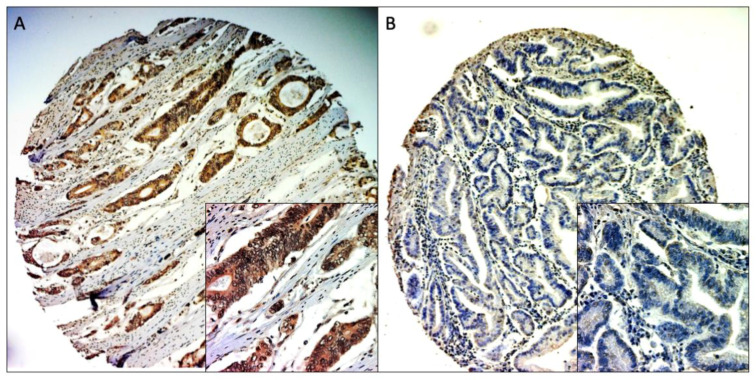
Representative MACC1 IHC staining of TMA cores. (**A**) positive staining (100× and 400× magnitude), (**B**) negative staining (100× and 400× magnitude).

**Figure 2 cancers-14-01773-f002:**
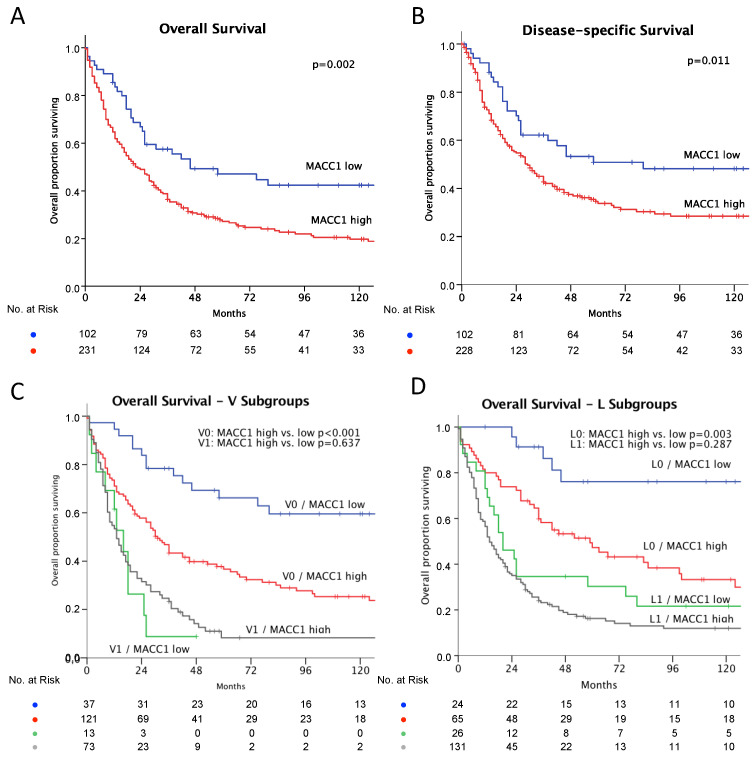
Kaplan–Meier plots of survival, MACC1-low: blue and MACC1-high: red. Significance calculated by log-rank test. (**A**) Overall survival: MACC1-low, better survival, *p* = 0.002; (**B**) Disease specific survival: MACC-low better survival, *p* = 0.011. (**C**) Overall survival—subgroup impact of vein invasion. No vein invasion: blue and red as above; MACC-low, better survival, *p* < 0.001. With vein invasion: MACC1-low: green; MACC1-high: grey; difference not significant, *p* = 0.637. (**D**) Overall survival—subgroup impact of lymphatic vessel invasion. No lymphatic vessel invasion: blue and red as above; MACC-low, better survival, *p* = 0.003; With lymphatic vessel invasion: MACC1-low: green; MACC1-high: grey; difference not significant: *p* = 0.287.

**Figure 3 cancers-14-01773-f003:**
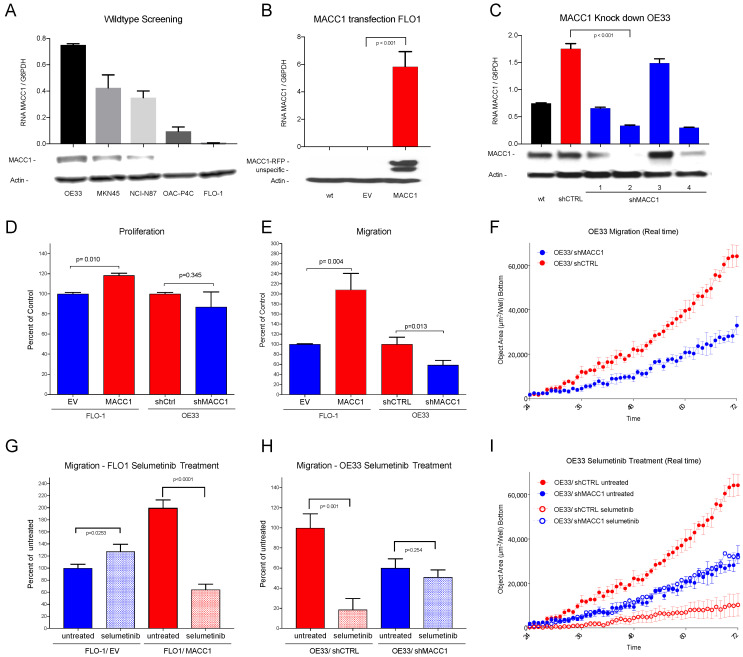
(**A**) Western blot and RNA expression of cell lines OE33, MKN45, NCI-N87, OAC-P4C and FLO-1: high MACC1 expression in OE33 cells, negligible MACC1 expression in FLO-1 cells. (**B**) Western blot and RNA expression of MACC1-transfected cell line FLO-1. wt = wild type, EV = empty vector and MACC1; Significant overexpression of MACC1 in FLO-1/MACC1 compared to FLO-1/EV (*p* < 0.001). (**C**) Western blot and RNA expression of different MACC1-downregulated OE33 subclones. wt = wild type, shCTRL = scrambled sh control, shMACC1 1–4 = 4 different MACC1 small hairpin vectors. (**D**) The FLO-1/MACC1 clone shows higher proliferation over 24 h compared to FLO-1/EV (*p* = 0.010). OE33/shMACC1 showed no statistically significant difference (*p* = 0.345) in comparison to OE33/shCTRL. (**E**) In the migration assay, over 72 h FLO-1/MACC1 cells migrate significantly more than FLO-1/EV cells (*p* = 0.004), and OE33/shMACC1 cells significantly less than OE33/shCTRL cells (*p* = 0.013). (**F**) Effects of MACC1 inhibition on migration in a real time plot over 72 h for OE33/shMACC1-negative (blue) and MACC1-positive OE33/shCTRL (red) cells. (**G**) Treatment with selumetinib (light colors) decreases migration of FLO-1/MACC1 cells (red) compared to untreated cells (dark colors) significantly (*p* < 0.001). (**H**) Treatment with selumetinib (light colors) decreases migration of MACC1-positive OE33/shCTRL cells significantly compared to migration in untreated cells (*p* = 0.001) (red). (**I**) Effects on migration by inhibition with selumetinib in a real time plot over 72 h: MACC1 knockdown OE33/shMACC1 cells in blue and MACC1-positive OE33/shCTRL in red. Selumetinib-treated cells are shown in light and untreated in dark colors. Significant effects on selumetinib treatment are only shown in the MACC1-expressing OE33/shCTRL cells.

**Figure 4 cancers-14-01773-f004:**
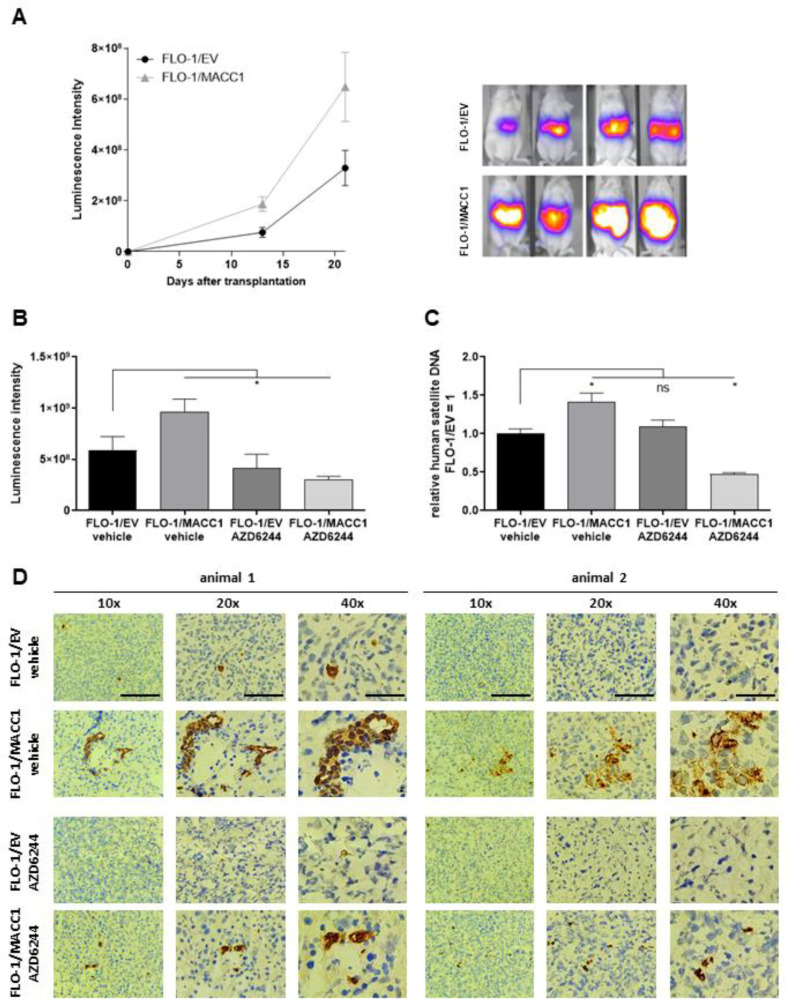
Effect of selumetinib (AZD6244) on MACC1-induced metastasis formation in a mouse xenograft model. (**A**) NOG mice were xenotransplanted with FLO-1/EV/Luc and FLO-1/MACC1/Luc cells by intrasplenal injection (*n* = 6 per group). Metastasis formation in the liver was monitored by BLI for 21 days (left panel). Representative BLI overlays of the animals are shown in the right panel. In the next experiment, NOG mice (*n* = 10 per group) were xenotransplanted with FLO-1/EV/Luc and FLO-1/MACC/Luc. Mice were treated with either vehicle or AZD6244. (**B**–**D**) Metastasis formation in the liver was analyzed after three weeks by BLI (**B**) and human satellite DNA (**C**), and visualized by human CK19 staining of murine liver tissue (**D**). *: *p* < 0.05.

**Table 1 cancers-14-01773-t001:** Patient characteristics of the analyzed patient cohort and distribution of MACC1-high and MACC1-low tumors. Significance calculated by x2-Test.

	Total		MACC1 Low		Macc1 High		Exluced		*p*
	*n*	(%)	*n*	(%)	*n*	(%)	*n*	(%)	
Gender									
Female	131	(36.4)	14	(10.7)	82	(62.6)	35	(26.7)	0.083
Male	229	(63.6)	41	(17.9)	129	(56.3)	59	(25.8)	
Age Group									
<65 years	159	(44.2)	37	(23.9)	111	(69.8)	53	(33.3)	0.051
≥65 years	201	(55.8)	18	(9.0)	100	(49.8)	41	(20.4)	
Localization									
AEG	60	(83.3)	5	(8.3)	41	(68.3)	14	(23.3)	0.161
Stomach	300	(16.7)	50	(16.7)	170	(56.7)	80	(26.7)	
Tumor Stage									
T1	47	(13.1)	15	(31.9)	22	(46.8)	10	(21.3)	**0.001**
T2	140	(38.9)	25	(17.9)	81	(57.9)	34	(24.3)	
T3	122	(33.9)	10	(8.2)	86	(70.5)	26	(21.3)	
T4	32	(8.9)	5	(15.6)	22	(68.8)	5	(15.6)	
Tx	19	(5.3)	0	(0.0)	0	(0.0)	19	(100.0)	
Node Stage									
N0	105	(29.29	19	(18.1)	51	(48.6)	35	(33.3)	0.293
N1	108	(30.0)	18	(16.7)	65	(60.2)	25	(23.1)	
N2	71	(19.7)	10	(14.1)	45	(63.4)	16	(22.5)	
N3	75	(20.8)	8	(10.7)	50	(66.7)	17	(22.7)	
Nx	1	(0.3)	-	-	-	-	-	-	
Distant Metastasis									
M0	245	(68.1)	48	(19.6)	149	(60.8)	48	(19.6)	**0.012**
M1	115	(31.9)	7	(6.1)	62	(53.9)	46	(40.0)	
Lymphatic Vessel Invasion									
L0	115	(31.9)	24	(20.9)	66	(57.4)	25	(21.7)	0.07
L1	206	(57.2)	26	(12.6)	131	(63.6)	49	(23.8)	
Unknown	39	(10.8)	-	-	-	-	-	-	
Vein Invasion									
V0	201	(55.8)	37	(18.4)	122	(60.7)	42	(20.9)	0.139
V1	114	(31.7)	13	(11.4)	73	(64.0)	28	(24.6)	
Unknown	45	(12.5)	-	-	-	-	-	-	
Grading									
G1	1	(0.3)	1	(100.0)	0	(0.0)	0	(0.0)	**0.001**
G2	90	(25.0)	23	(25.6)	48	(53.3)	19	(21.1)	
G3	266	(73.9)	29	(10.9)	71	(26.7)	166	(62.4)	
Unknown	3	(0.8)	-	-	-	-	-	-	
Lauren Classification									
Intestinal	122	(33.9)	26	(21.3)	69	(56.6)	27	(22.1)	**0.048**
Diffuse	190	(52.8)	19	(10.0)	114	(60.0)	57	(30.0)	
Mixed	45	(12.5)	8	(17.8)	27	(60.0)	10	(22.2)	
Unknown	3	(0.8)	-	-	-	-	-	-	
Missmatch Repair System									
proficient	278	(77.2)	49	(17.6)	183	(65.8)	46	(16.5)	**<0.0001**
deficient	39	(10.8)	5	(12.8)	35	(89.7)	9	(23.1)	
unkown	43	(11.9)	-	-	-	-	-	-	

**Table 2 cancers-14-01773-t002:** Univariate and multivariate analysis of patient survival dependent on pathomorphological criteria. Univariate analysis performed by log-rank and multivariate analysis by Cox’s regression.

	Univariate Analysis		Multivariate Analysis			
	Median Survival Months (95% CI)	*p*	Odds Ratio	Low	High	*p*
Tumor Stage						
T1	181.19 (164.7–197.7)	**<0.0001**	1.798	**1.467**	**2.203**	**<0.0001**
T2	79.72 (67.2–92.2)					
T3	32.76 (23.5–42.1)					
T4	14.13 (9.3–19.0)					
Node Stage						
N0	135.79 (121.3–150.3)	**<0.0001**	1.595	**1.353**	**1.879**	**<0.0001**
N1	64.32 (51.6–77.0)					
N2	24.06 (16.6–31.6)					
N3	17.92 (10.8–25.0)					
Nx	13.00 (13.0–13.0)					
Lymphatic Vessel Invasion						
L0	119.68 (103.3–136.0)	**<0.0001**	1.004	0.774	1.302	0.976
L1	43.27 (34.9–51.6)					
Lx	32.7 (15.9–49.1)					
Vein Invasion						
V0	99.00 (86.2–111.8)	**<0.0001**	1.003	0.778	1.292	0.984
V1	34.56 (24.7–44.4)					
Vx	32.22 (16.6–47.8)					
Grading						
1	-	**0.038**	0.828	0.551	1.243	0.791
2	88.71 (72.7–104.7)					
3	76.4 (65.8–87.0)					
Lauren Classification						
intestinal	95.97 (82.0–109.9)	**0.002**	1.038	0.786	1.373	0.791
diffuse	69.83 (56.8–82.9)					
mixed	69.40 (51.4–87.4)					
Mismatch Repair System						
proficient	47.22 (28.1–66.3)	**0.025**	0.759	0.494	1.165	0.207
deficient	79.77 (70.2–89.4)					
MACC1						
low	87.06 (66.5–107.6)	**0.002**	1.513	**1.013**	**2.258**	**0.043**
high	54.20 (45.0–63.4)					

## Data Availability

Data will be made available by the corresponding author on reasonable request.

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
