# Peer review of "Inhibition of MACC1-Induced Metastasis in Esophageal and Gastric Adenocarcinomas"

_cancers, 2022, doi:10.3390/cancers14071773_

Round 1
Reviewer 1 Report
In this paper the authors represents that MACC1 is a negative prognostic marker in esophageal and gastric adenocarcinomas in a large Caucasian cohort. They demonstrate that MACC1 is expressed in 79% of the investigated samples and its presence is coupled with reduced overall survival even in patients with good prognosis. In the in vitro analysis they investigated MACC1 expression in five cell lines and generated a cell line with MACC1 overexpression and a MACC1 knock-down cell line. They found that MACC1 expression does not influence cell proliferation but is correlated with cell migration. Inhibition of MEK1 with selumetinib also reduced cell migration only in MACC1 expressing cells. In an in vivo xenograft model using a control and a MACC1 overexpressing cell line pair they found that MACC1 expression increased metastasis formation and this was reduced by selumetinib treatment.
Comments to authors:
Major:
- Based on the cell line RNA and protein data the expression level of MACC1 is highly variable rather than being present or not. Is there a similar variability present among the tissue samples? Please provide the distribution of the IRS scores of the patient cohort and explain how was IRS 5 chosen as cut off?
- The authors argue that MACC1 expression did not significantly influenced cell proliferation in the two cell line models. However, cell proliferation was analyzed only in a 24-hour long period even though most cancer cells have a doubling time around 24 hours. In order to draw a conclusion about the proliferation effect a longer (48 or 72 hours) measurement is necessary.
- The concentration used from selumetinib both for the cell line experiments (10 microM) and for the animal experiments (50mg/kg) are quite high. Please discuss how the applied concentrations are related to the clinicaly used dosage (75 mg administered orally twice https://clinicaltrials.gov/ct2/show/NCT02448290) and why not the more efficient MEK1 inhibitor trametinib was used for the experiments. It would be nice to know the selumetinib sensitivity of the two used cell lines.
Author Response
Dear Professor Mok,
We appreciate that you give us the possibility to revise our manuscript and we thank the reviewers for their valuable comments. Based on these comments we herewith respond to all questions raised in a point-by-point manner, as follows:
Reviewer 1:
- Based on the cell line RNA and protein data the expression level of MACC1 is highly variable rather than being present or not. Is there a similar variability present among the tissue samples? Please provide the distribution of the IRS scores of the patient cohort and explain how was IRS 5 chosen as cut off?
Answer: We thank the reviewer for this comment.
As suggested, we analyzed the distribution of IRS (immunoreactivity score) across the cohort. In Figure 1 it can be clearly seen that there are two groups forming here, those with 0-4 and those with 6-12 IRS.
In addition, we calculated the survival probability depending on the IRS (Figure 2). Patients with an IRS of 6 or higher have a similar bad prognosis compared to patients with an IRS below 5. A better survival correlates with lower MACC1 IRS. Patients with an IRS of 0 have the best prognosis.
Based on these two analyses, the decision for cut-off value of 5 was made.
Additionally, ROC curve analysis showed, that the IRS value of 5 has the highest Youden Index and is therefore the most suitable cut-off value. The IRS cut-off value 5 reaches a sensitivity of 76.6% and a specificity of 64.2% (Youden Index. 0.408) (Figure 3 and Table 1).
Looking at MACC1 expression in cell lines and tissue samples, the variability looks similar. Nevertheless, it must be considered that the investigation of the homogeneous cell lines is a purely quantitative analysis of the expression by Western plot and RT-PCR. The analysis of the tissue samples by immunoreactivity score analyses both the strength of expression and the inhomogeneity of expression.
Figure 1: Distribution of the IRS across the cohort
Figure 2: Overall Survival depending on the IRS
Figure 3: ROC Curve for IRS cut-off analysis (AUC 0.716 [0.656 – 0.776])
|
Positive if Greater than or Equal to |
Sensitivity |
1- Specificity |
Specificity |
Youden Index |
|
-1 |
1 |
1 |
0 |
0 |
|
1 |
0,869 |
0,500 |
0,500 |
0,369 |
|
3 |
0,864 |
0,483 |
0,517 |
0,381 |
|
5 |
0,766 |
0,358 |
0,642 |
0,408 |
|
7 |
0,729 |
0,342 |
0,658 |
0,387 |
|
8,5 |
0,421 |
0,217 |
0,783 |
0,204 |
|
10,5 |
0,411 |
0,200 |
0,800 |
0,211 |
|
13 |
0 |
0 |
1 |
0 |
Table 1: Sensitivity, specificity and Youden Index for different IRS cut-off values.
- The authors argue that MACC1 expression did not significantly influenced cell proliferation in the two cell line models. However, cell proliferation was analyzed only in a 24-hour long period even though most cancer cells have a doubling time around 24 hours. In order to draw a conclusion about the proliferation effect a longer (48 or 72 hours) measurement is necessary.
Answer: We thank the reviewer for raising this point. However, to respond to this question appropriately, we currently can provide only the results from proliferation assays for one cell line model (FLO1-EV vs FLO1-MACC1) over the time period of 72 h. This assay was with no statistically significant difference in cell proliferation between FLO1-EV vs FLO1-MACC1 (see Figure 4).
Figure 4: Proliferation of FLO1 EV and FLO1 MACC1 over 72 h. Analysis of in vitro proliferation assay.
Unfortunately, the short revision period of 10 days limits the generation of additional data sets for proliferation in further tumor cell lines.
- The concentration used from selumetinib both for the cell line experiments (10 microM) and for the animal experiments (50 mg/kg) are quite high. Please discuss how the applied concentrations are related to the clinically used dosage (75 mg administered orally twice https://clinicaltrials.gov/ct2/show/NCT02448290) and why not the more efficient MEK1 inhibitor trametinib was used for the experiments. It would be nice to know the selumetinib sensitivity of the two used cell lines.
Answer: We thank the reviewer for this question. The concept of MACC1 inhibition was originally developed by our group for the treatment of MACC1 positive colorectal cancers and was recently published (Kobelt et al. Oncogene 2021). In this study we transferred our experience with MACC1 inhibition by selumetinib on AGE/S. Therefore, we chose the same doses for in vitro and in vivo experiments, and which was also used by other authors (Huynh et al. Mol Cancer Therapy. 2007). Further, we knew that the selected selumetinib concentrations of 10 µM has no cytotoxic effect in vitro and showed no reduction of migration in the MACC1 negative clones (FLO1-EV and OEshMACC1).
Regarding the different dosages in the context of xenografted mice treatment in vivo and the clinically used dosages, the FDA recommends multiplying the dosage by a factor of 0.08 when transferring from mouse to human (Niar et al. Journal of basic and clinical pharmacy 2016). According to this calculation, the dosage of 50 mg selumetinib /kg mouse used would equal a dosage of 4 mg/kg in humans. The clinical dose of 75mg twice daily corresponds to a dose of 2x1mg per kg assuming a patient weight of 75kg. Using the aforementioned calculation, this dose equals then 12.5 mg/kg for treatment of mice. In our study, we refer again to the work of Huynh et al. Here, in which a dosage of 12 mg/kg selumetinib showed only a low level of effectiveness. At 100 mg/kg, a very good effectiveness was seen, however combined with an increased toxicity. Thus, we used in our study the dosage of 50 mg/kg as an effective dosage with low toxicity for the mice.
Overall, we are considering here a pre-clinical model. The possibility of MEK1 inhibition to inhibit MACC1-induced effects in AGE/S was examined. Further, extended in vivo studies might then reveal the ideal dosage in vivo, being thereafter translated to use in humans. The use of trametinib is an interesting suggestion and might well be adapted to its use in forthcoming studies
References:
- Kobelt, D., Perez-Hernandez, D., Fleuter, C., Dahlmann, M., Zincke, F., Smith, J., ... & Stein, U. (2021). The newly identified MEK1 tyrosine phosphorylation target MACC1 is druggable by approved MEK1 inhibitors to restrict colorectal cancer metastasis. Oncogene, 40(34), 5286-5301.
- Huynh H, Soo KC, Chow PK, Tran E. Targeted inhibition of the extracellular signal-regulated kinase kinase pathway with AZD6244 (ARRY-142886) in the treatment of hepatocellular carcinoma. Mol Cancer Ther. 2007 Jan;6(1):138-46..
- Nair, Anroop B.; Jacob, Shery. (2016). A simple practice guide for dose conversion between animals and human. Journal of basic and clinical pharmacy, 7. Jg., Nr. 2, S. 27.
Reviewer 2 Report
Previous studies in colorectal cancer have shown that MACC1 expression (which is associated with poor prognosis) induces cell migration in vitro and metastasis formation in vivo, and these effects can be inhibited by selumetinib. In this study, the authors have shown that the same is also true for gastric/esophageal cancer. The results are interesting and may have clinical relevance. I recommend the following edits before publication.
Major Issues:
[1] Figure 4 is missing. In the pdf I reviewed, the caption is there, but the figure is not.
[2] Abstract says 266 patients were analyzed (line 36), while result says 360 patients were analyzed (line 249). Please fix this inconsistency.
[3] Throughout the paper, 10^x wrongly displays as 10x (e.g. "105" instead of "10^5" in lines 152 and 170, "106" instead of "10^6" in line 217, "X2" instead of "X^2" in line 241, "108" instead of "10^8" in lines 353, 354, 365, 366, 367, 368, 370, 371).
Minor Issues:
[4] In Figures 3G and 3H, please specify p-values for untreated vs selumetinib in FLO-1/EV and OE33/shMACC1 respectively. If the difference is statistically significant, please briefly explain/speculate why.
[5] If possible, please use a higher resolution version of Figures 3F and 3I (the legends are blurry and hard to read in the current version).
[6] In line 306/307, please explain/speculate briefly why the control clone had higher MACC1 expression than the wildtype.
[7] In line 272/273: (i) please report median survival in each group (instead of mean survival), (ii) please specify Hazard ratio with 95% confidence interval (in addition to p values), (iii) please report both overall survival result and disease-specific survival result (since you showed both in Figures).
[8] In line 274, please change "OR 1.51" to "OR 1.51 [... - ...]", i.e. specify the 95% confidence interval.
[9] In line 277-279: (i) please report median survival in each group (instead of mean survival), (ii) please specify Hazard ratio with 95% confidence interval (in addition to p values). Same for lines 425/426.
[10] If the journal allows it, please break up the abstract into Introduction, Methods, Results, Conclusion. If the journal requires an unstructured abstract instead, please reorganize the current abstract so that the content flows uninterrupted (e.g. "MACC1 is an independent negative prognostic marker in AGE/S patients." should be after "Expression was correlated with survival and morphological characteristics." and before "To analyze the role of MACC1 in vitro the cell lines FLO-1 and OE33 were lentivirally manipulated." and so on).
Author Response
Dear Professor Mok,
We appreciate that you give us the possibility to revise our manuscript and we thank the reviewers for their valuable comments. Based on these comments we herewith respond to all questions raised in a point-by-point manner, as follows:
Reviewer 2
Major Issues:
[1] Figure 4 is missing. In the pdf I reviewed, the caption is there, but the figure is not.
We apologize for the inconvenience. Figure 4 can be found on page 14 in the submitted version. We inserted Figure 4 in the revised manuscript again to solve the technical issue.
[2] Abstract says 266 patients were analyzed (line 36), while result says 360 patients were analyzed (line 249). Please fix this inconsistency.
Answer: We address this question in the results part MACC1 expression: “ Due to the procedure of cutting and staining, some samples could not be used for the evaluation. Samples with insufficiently representative tumor tissue were also excluded from the evaluation. Cores with representative tumor material and evaluable staining were available in 266 of 360 samples (73.9%)…”.
To avoid misunderstandings, we corrected the abstract to: 266 Samples of 360 AGE/S patients were analyzed for MACC1 expression.
[3] Throughout the paper, 10^x wrongly displays as 10x (e.g. "105" instead of "10^5" in lines 152 and 170, "106" instead of "10^6" in line 217, "X2" instead of "X^2" in line 241, "108" instead of "10^8" in lines 353, 354, 365, 366, 367, 368, 370, 371).
Answer: We thank the reviewer for this correction. This formatting issue was corrected throughout the entire manuscript.
Minor Issues:
[4] In Figures 3G and 3H, please specify p-values for untreated vs selumetinib in FLO-1/EV and OE33/shMACC1 respectively. If the difference is statistically significant, please briefly explain/speculate why.
Answer: We thank the reviewer for this valuable hint. As suggested, we included p-values in Figure 3G and 3H. In Figure 3G there is a significant increase in migration of cells without MACC1 expression. This is based on a sufficiently large N, that renders even small differences statistically significant. Further, this increase is rather small and is outperformed by the MACC1-mediated increase in migration and the reduction in the MACC1-mediated migration by selumetinib. The data show, that selumetinib does not reduce migration of the MACC1-negative FLO1 and that MACC1 induces migration in this cell line that can be inhibited by selumetinib.
[5] If possible, please use a higher resolution version of Figures 3F and 3I (the legends are blurry and hard to read in the current version).
Answer: We thank the reviewer for this suggestion to improve the visual quality of the manuscript. The legends of 3F and I have been enlarged and the resolution of the figure has been increased.
[6] In line 306/307, please explain/speculate briefly why the control clone had higher MACC1 expression than the wildtype.
We thank the reviewer for this discussion. The empty vector control clone was transduced and selected afterwards. During the selection procedure it is possible that a subclone was generated that expresses slightly more (or less) of any mRNA and protein. It differs from the wildtype by handling (transduction, selection etc) and somewhat by age (cell divisions). Therefore, for all comparisons the empty vector clone was used, as this clone was generated in parallel to the MACC1 clone.
[7] In line 272/273: (i) please report median survival in each group (instead of mean survival), (ii) please specify Hazard ratio with 95% confidence interval (in addition to p values), (iii) please report both overall survival result and disease-specific survival result (since you showed both in Figures).
Answer: We thank the reviewer for pointing this out: Survival reports were changed in median survival and the 95% confidence interval were included. Additionally, disease-specific survival results were reported.
[8] In line 274, please change "OR 1.51" to "OR 1.51 [... - ...]", i.e. specify the 95% confidence interval.
Answer: We thank the reviewer for pointing this out: OR with the confidence interval is now included.
[9] In line 277-279: (i) please report median survival in each group (instead of mean survival), (ii) please specify Hazard ratio with 95% confidence interval (in addition to p values). Same for lines 425/426.
Answer: We thank the reviewer for pointing this out: all survival reports were changed in median survival and the 95% confidence interval were included.
[10] If the journal allows it, please break up the abstract into Introduction, Methods, Results, Conclusion. If the journal requires an unstructured abstract instead, please reorganize the current abstract so that the content flows uninterrupted (e.g. "MACC1 is an independent negative prognostic marker in AGE/S patients." should be after "Expression was correlated with survival and morphological characteristics." and before "To analyze the role of MACC1 in vitro the cell lines FLO-1 and OE33 were lentivirally manipulated." and so on).
Answer: Thank you for this valuable comment. The journal requires an unstructured abstract. As suggested by the reviewer, we have therefore reorganized the abstract and improved the flow of reading:
Abstract:
Esophageal and Gastric Adenocarcinomas (AGE/S) are characterized by early metastasis and poor survival. MACC1 (Metastasis Associated in Colon Cancer 1) acts in colon cancer as a metastasis inducer, linked to reduced survival. This project illuminates the role and potential of inhibition of MACC1 in AGE/S.
Using 266 of 360 TMAs and survival data of AGE/S patients we confirm the value of MACC1 as an independent negative prognostic marker in AGE/S patients. MACC1 gene expression correlated with survival and morphological characteristics. In vitro analysis of lentivirally MACC1 manipulated subclones of FLO-1 and OE33 showed enhanced migration induced by MACC1 in both cell line models, which could be inhibited by the MEK1 inhibitor selumetinib. In vivo, the efficacy of selumetinib on tumor growth and metastasis of MACC1-overexpressing FLO-1 cells xenografted intrasplenically in NOG mice was tested. Mice with high MACC1 expressing cells developed faster and larger distant metastases. Treatment with selumetinib led to a significant reduction of metastasis exclusively in the MACC1 positive xenografts.
Round 2
Reviewer 1 Report
- The authors´answer regarding the cut off is sufficient, please include this argument either in the methods section or as supplementary information.
- The authors refer to a data "Proliferation of FLO1 EV and FLO1 MACC1 over 72 h", however I couldn´t find the figure either in the manuscript file or here in the answer.
Reviewer 2 Report
The authors have satisfactorily addressed my comments.
Author Response
There were no more open questions in the second review.